# Incidence Rate and Predictors of Globus Pallidus Necrosis after Charcoal Burning Suicide

**DOI:** 10.3390/ijerph16224426

**Published:** 2019-11-12

**Authors:** Chung-Hsuan Ku, Wen-Hung Huang, Ching-Wei Hsu, Yu-Chin Chen, Yi-Chou Hou, I-Kuan Wang, Hsiang-Hsi Hong, Yen-Li Wang, Cheng-Hao Weng, Tzung-Hai Yen

**Affiliations:** 1Department of Nephrology, Clinical Poison Center, Kidney Research Center, Center for Tissue Engineering, Chang Gung Memorial Hospital, Chang Gung University, Linkou 333, Taiwan; koshelton@cgmh.org.tw (C.-H.K.); williammedia@yahoo.com.tw (W.-H.H.); wei2838@gmail.com (C.-W.H.); 2Center for Traditional Chinese Medicine, Chang Gung Memorial Hospital, Taoyuan 333, Taiwan; yuchin.me@gmail.com; 3Department of Psychology, University of Arizona, Tuscon, AZ 85721, USA; 4Division of Nephrology, Department of Internal Medicine, Cardinal Tien Hospital, School of Medicine, Fu-Jen Catholic University, New Taipei City 23155, Taiwan; athletics910@gmail.com; 5Department of Nephrology, China Medical University Hospital, China Medical University, Taichung 40402, Taiwan; ikwang@mail.cmuh.org.tw; 6Department of Periodontics, Chang Gung Memorial Hospital, Chang Gung University, Linkou 333, Taiwan; f1214@adm.cgmh.org.tw; 7Department of Periodontics, Chang Gung Memorial Hospital, Chang Gung University, Taoyuan 333, Taiwan; Wanglacy519@yahoo.com.tw

**Keywords:** charcoal burning, suicide, globus pallidus necrosis, mortality

## Abstract

*Objective*: This study examined predictors of globus pallidus necrosis as there was a paucity of literature of globus pallidus necrosis resulted from carbon monoxide poisoning after charcoal burning suicide. *Methods*: A total of 67 patients who had attempted charcoal burning suicide were recruited and stratified into two subgroups based on either presence (*n* = 40) or absence (*n* = 27) of globus pallidus necrosis. Demographic, clinical, laboratory, and radiographic data were obtained for cross-sectional analysis. All patients were followed to investigate the risks for mortality. *Results*: The patients aged 36.8 ± 11.1 years (67.2%) were male. Patients with globus pallidus necrosis were younger (*p* = 0.044) and had less hypertension (*p* = 0.015) than patients without globus pallidus necrosis. Furthermore, patients with globus pallidus necrosis suffered from severer medical complications, i.e., fever (*p* = 0.008), acute myocardial injury (*p* = 0.022), acute rhabdomyolysis (*p* = 0.022), and neuropsychiatric symptoms (*p* < 0.001) than patients without globus pallidus necrosis. Moreover, patients with globus pallidus necrosis received less hyperbaric oxygen therapy than without necrosis (*p* = 0.024). Two patients (3.0%) died on arrival. In a multivariable regression model, it was revealed that acute myocardial injury (odds ratio 4.6, confidence interval 1.1–18.9, *p* = 0.034) and neuropsychiatric symptoms (odds ratio 8.0, confidence interval 2.0–31.4, *p* = 0.003), decreased blood bicarbonate level (odds ratio 0.8, confidence interval 0.7–1.0, *p* = 0.032), and younger age (odds ratio 0.9, confidence interval 0.9–1.0, *p* = 0.038) were significant predictors for globus pallidus necrosis. *Conclusion*: Although patients who had attempted charcoal burning suicide had a low mortality rate (3.0%), globus pallidus necrosis was not uncommon (59.7%) in this population. Further studies are warranted.

## 1. Introduction

Charcoal burning suicide is a serious public health problem in Asia. Since the first case of suicide by burning charcoal in 1998 in Hong Kong, cases have increased in many Far East areas, such as Taiwan, Japan, Korea, and China. In Taiwan, 32 people committed suicide by burning charcoal in 1998, but 1346 cases were noted in 2005 [1]. This endemic of charcoal burning suicide was thought to be promoted by media because this way of suicide is always described as painless and easy [2]. In Taiwan, the number of charcoal burning suicide attempts was the second biggest method of all suicides, and comprised 33.5% of all suicide related deaths [3].

Carbon monoxide (CO) poisoning is the main cause of mortality and morbidity after charcoal burning suicide. The most common brain lesion is globus pallidus necrosis [4,5]. Symmetric hypodensity over globus pallidus can be found in computed tomography (CT). The medial portion of the globus pallidus shows low intensity on T1-weighted images but high intensity on T2-weighted images on magnetic resonance imaging (MRI) [6]. The pathophysiology of globus pallidus necrosis remains unclear. One study revealed that learning and memory deficit was noted in rats exposed to CO, and there were higher glutamate and hydroxide levels in these rats [7]. Another study noted a hypotensive effect in the watershed territory of the arterial supply and hypoxia due to decreased arterial supplement or CO binding to globus pallidus, which is an iron-rich area [6]. Hypotension or shock status either due to sepsis, metabolic acidosis, or myocardial infarction could contribute to poorer brain perfusion. Because of the higher affinity of CO to hemoglobin over oxygen, this hypoxic effect can be systemic and produces multi-organ failure [8].

Risk factors for mortality after CO poisoning have been reported in the literature. Hypothermia, respiratory failure, renal failure, or coma indicates poor prognosis [9]. Acute kidney injury has been revealed as another risk factor for short-term mortality after charcoal burning suicide [8]. However, few reports have analyzed the outcomes of charcoal burning suicide in the context of globus pallidus necrosis. Therefore, the objective of this study was to examine predictors of globus pallidus necrosis in this charcoal burning population.

## 2. Methods

This retrospective observational study followed the Declaration of Helsinki and was approved by the Medical Ethics Committee of Chang Gung Memorial Hospital (Institutional Review Board code number: 201701109B0).

### 2.1. Patients

Between 2002 and 2013, a total of 67 patients who had attempted charcoal burning suicide were admitted to Chang Gung Memorial hospital, Linkou, Taiwan. All patients received psychiatric consultation-liaison services. Demographic data such as age, gender, underlying disease, personal habits, and previous suicide attempt history were recorded. Clinical data such as fever, acute respiratory failure, acute myocardial infarction, acute hepatic injury, acute kidney injury, rhabdomyolysis, neuropsychiatric symptoms, stroke, shock, out-of-hospital cardiac arrest, etc. were analyzed. Laboratory data such as blood tests and radiographic examinations were obtained. The carboxyhemoglobin levels were measured at admission. Treatment modalities such as oxygen therapy, hyperbaric oxygen therapy, and mortality were collected.

### 2.2. Oxygen Therapy

None of the patients received oxygen therapy prior to admission. After admission, patients were treated with oxygen therapy via a non-rebreather facemask or hyperbaric oxygen (HBO) therapy. There was no standard indication for HBO treatment [8]. Nevertheless, the absolute contraindication for HBO was an untreated pneumothorax. Moreover, the relative contraindications comprised asthma, chronic obstructive pulmonary disease, seizure, fever, active infection, active malignancy, claustrophobia, pregnancy, and patients who needed intensive medical care [8].

### 2.3. Inclusion and Exclusion Criteria

All patients who had attempted charcoal burning suicide and had received brain image analysis were enrolled into this study, but patients with CO intoxication due to other etiologies were excluded from the analysis. There was no routine use of brain imaging to aid in the clinical management of patients, but 67 out of 126 patients received brain image analysis.

### 2.4. Definition of Globus Pallidus Necrosis

The diagnosis of globus pallidus necrosis was confirmed by radiographic imaging studies. The typical findings were low intensity on T1-weighted images but high intensity on T2-weighted images over the bilateral globus pallidus area in MRI, or symmetric hypodensity over globus pallidus in CT.

### 2.5. Definition of Neuropsychiatric Symptoms

Neuropsychiatric symptoms were defined as a new onset of cognitive and personality changes, dementia, psychosis, Parkinsonism, amnesia, depression, and incontinence after charcoal burning suicide.

### 2.6. Statistical Analysis

Normality of distribution and equality of standard deviation was tested in all variables before analysis. Continuous variables were expressed as means ± standard deviations whereas categorical variables were expressed as numbers (percentages). Non-normal distribution data were presented as median (interquartile range). Comparisons between two groups were performed using the Student’s T test for continuous variables and the Chi-square or Fisher’s exact test for categorical variables. A univariable logistic regression analysis was performed to compare the frequency of possible risk factors associated with globus pallidus necrosis. To control for confounding factors, a multivariable logistic regression analyses was performed to analyze the significant factors on univariable analysis that met the assumptions of a proportional hazard model. All statistical tests were two-tailed, with *p* values <0.05 being considered statistically significant. Data were analyzed with IBM SPSS Statistics Version 20 (IBM Corporation, Armonk, NY, USA).

## 3. Results

Table 1 describes the baseline demographic characteristics of 67 patients who had attempted charcoal burning suicide, stratified according to either presence (*n* = 40) or absence (*n* = 27) of globus pallidus necrosis. The patients aged 36.8 ± 11.1 years (67.2%) were male. It was found that patients with globus pallidus necrosis were younger (34.5 ± 9.6 vs. 40.1 ± 12.4, *p* = 0.044) and had less hypertension (2.5% vs. 22.2%, *p* = 0.015) than patients without globus pallidus necrosis. Otherwise, there were no significant differences in other baseline variables (*p* > 0.05).

As shown in Table 1, patients with globus pallidus necrosis suffered from severer medical complications when they were admitted to the hospital, i.e., fever (47.5% vs. 14.8%, *p* = 0.008), acute myocardial injury (52.5% vs. 22.2%, *p* = 0.022), acute rhabdomyolysis (52.5% vs. 22.2%, *p* = 0.022) and neuropsychiatric symptoms (70.0% vs. 22.2%, *p* < 0.001) than patients without globus pallidus necrosis. Additionally, patients with globus pallidus necrosis had poorer blood test results, i.e., aspartate transaminase (119 U/L vs. 39.0 U/L, *p* = 0.001), creatinine kinase (4610 U/L vs. 303 U/L, *p* = 0.020), creatine kinase–MB (19.2 U/L vs. 4.1 U/L, *p* = 0.014), creatinine (1.37 mg/dL vs. 0.9 mg/dL, *p* = 0.022), troponin I (1.73 vs. 0.14, *p* < 0.001), and bicarbonate (19.1 ± 4.5 mg/dL vs. 22.1 ± 4.1 mg/dL, *p* = 0.007) than patients without globus pallidus necrosis. However, there was no significant difference in blood carboxyhemoglobin levels between both groups (18.4% ± 3.2% vs. 26.7% ± 3.5%, *p* = 0.086). All patients received oxygen therapy using non-rebreather facemasks. Nevertheless, patients with globus pallidus necrosis received less hyperbaric oxygen therapy than patients without globus pallidus necrosis (30.0% vs. 59.3%, *p* = 0.024).

The results of univariable and multivariable logistic regression analysis are shown in Table 2 and Table 3. In a multivariable regression model (Table 3), it was disclosed that acute myocardial injury (odds ratio 4.6, confidence interval 1.1–18.9, *p* = 0.034) and neuropsychiatric symptoms (odds ratio 8.0, confidence interval 2.0–31.4, *p* = 0.003) were significant risk factors for globus pallidus necrosis. In contrast, bicarbonate level (odds ratio 0.8, confidence interval 0.7–1.0, *p* = 0.032), and age (odds ratio 0.9, confidence interval 0.9–1.0, *p* = 0.038) were protective factors associated with lower risk of globus pallidus necrosis.

## 4. Discussion

Our analytical data (Table 1) revealed that although patients who had attempted charcoal burning suicide had a low mortality rate (3.0%), globus pallidus necrosis was not uncommon (59.7%). There were some studies in the literature that had discussed the relationship between CO intoxication and globus pallidus necrosis (Table 4). Most of the studies were from Asian countries, and the incidence rates of globus pallidus necrosis after CO intoxication ranged between 18.7% and 100% [4,10,11,12,13,14,15,16,17,18,19,20]. Nevertheless, the sources of CO were not mentioned in many of the studies. A large registry-based study from Korea [11] revealed that globus pallidus necrosis developed in 19.9% of patients, but the source of CO was not described, and the mortality rate was not reported. Another Korean study [12] in patients with charcoal burning reported an 18.7% rate of globus pallidus necrosis and a 0% mortality rate. Compared with the above two studies, it seemed that our patients suffered a higher incidence rate of globus pallidus necrosis (59.7%) and had a slightly higher mortality rate (3.0%).

A total of 56.7% patients had mood disorder, and 19.4% has past suicide attempt history (Table 1). Chronic alcohol consumption (49.3%) and a smoking habit (55.2%) were noted in approximately half of patients. This characteristic of charcoal burning suicide patients was similar to other studies based in Korea, Hong Kong, and Japan [21,22,23,24]. An increasing trend in charcoal burning suicide was noted in Asian countries [23]. Mass media reporting plays an important role in the increased incidences of charcoal burning suicides [25,26,27]. The higher incidence rate is believed to be due to the traditional perception of the ethnic Chinese group who believe that a person should die with a complete body. The media portrayal of charcoal burning suicide as painless and easy, with no body disfigurement, contributes to the rapid spread of news in the era of globalization. 

In a multivariable regression model (Table 4), it was noted that neuropsychiatric symptoms (*p* = 0.003) and acute myocardial injury (*p* = 0.034) were powerful predictors of globus pallidus necrosis. Reduction of oxygen delivery along with mitochondrial oxidative phosphorylation could lead to ischemic and hypoxic brain injury in CO intoxication patients [28]. These patients commonly have neurologic symptom that include headache, dizziness, loss of consciousness, nausea, vomiting, etc. [29]. Delayed neurologic sequels, like depression, anxiety, memory symptoms, and motor deficits were also reported [29]. The incidence rate of neuropsychiatric symptoms in our study was 50.7% in all patients and was higher in patients with globus pallidus necrosis (70.0%, Table 1). According to a study [11], the incidence rate of globus pallidus lesions in CO intoxication patients was 19.9%, less than the 59.7% in our study. The percentage of delayed neurologic sequel was not different between the presence and absence of globus pallidus lesion (78.6% vs. 72.4%) [11]. Another study from Hong-Kong revealed that delayed neurologic sequel happened in 7.5% of patients with CO poisoning but 57.1% in the subgroup with globus pallidus lesions [30]. Therefore, the relationship between globus pallidus necrosis and acute neuropsychiatric symptoms or delayed neurologic sequel remains unclear and needs more study to evaluate their correlation.

There are some case reports noted about the correlation between acute myocardial injury and globus pallidus necrosis [25,31,32]. Only one registry-based cohort study showed that patients with CO poisoning have a higher risk of major adverse cardiovascular events [33]. The incidence rate of acute myocardial injury was 40.3% in our study (Table 1). Furthermore, patients with globus pallidus necrosis suffered a higher rate of acute myocardial injury than patients without globus pallidus necrosis (52.5% vs. 22.2%, *p* = 0.022). To our knowledge, there is a 200 times greater affinity of carbon monoxide to hemoglobin then to oxygen, so the human body would suffer from hypoxia damage in patients who had attempted suicide by charcoal burning. The organs affected the most by carbon monoxide are the central nervous system and the myocardium, whose organs need oxygen the most. The patients with globus pallidus necrosis were affected more by hypoxia, so these patients would also have myocardium injury. Also, a decrease in oxygen delivery would increase global oxygen demand and myocardial contractility, which would induce myocardial infarction in patients with underlying coronary artery disease [34]. Hypertension appeared to be a protective factor of globus pallidus necrosis in our study. While an elevation of blood pressure was noted, the increased blood perfusion for all organs, especially the brain, was also noted. Therefore, less hypoxia episodes, as well as globus pallidus necrosis, will be seen in the hypertension status. In support of this concept, a forensic pathology study [35] also indicated that globus pallidus necrosis is merely a consequence of cerebral hypoxia–ischemia from a wide variety of etiologies such as drug overdose, post-anesthesia, in children after cardiac surgery, hemolytic uremic syndrome, acute renal failure, metabolic disorders, viral infections, porphyria and cerebral arteriosclerosis, delayed hanging, and wasp stings, and it is not specific to any kind of injury.

HBO therapy is a reasonable therapy in acute CO poisoning. The mechanism involves increased oxygen content in the blood, so as to enhance the clearance of CO and to prevent hypoxemic brain injury. In our study, only 41.8% of patient who had attempted charcoal burning suicide received HBO therapy, and the figure was lower (30.0%) in patients with globus pallidus necrosis (Table 1). Previous studies also revealed that only 41.6% of patients who had attempted charcoal burning suicide [8] and 18.8% of patients with carbon monoxide poisoning [9] received HBO treatment. The correlation between HBO therapy with mortality was undetermined, and so far there is a lack of standard indication for HBO therapy. One retrospective study from the Nationwide Poisoning Database in Taiwan revealed a lower mortality rate in patients who received HBO therapy [36]. A single center retrospective study in Pittsburgh revealed that hyperbaric oxygen was associated with reduced acute and reduced one-year mortality [37]. However, no literature evidence indicates whether HBO therapy could improve globus pallidus necrosis or not. Although the percentage of HBO therapy was lower in our patients with globus pallidus necrosis (30.0%) than in patients without globus pallidus necrosis (59.3%, Table 1), HBO was not a significant predictor for globus pallidus necrosis after multivariable logistic regression analysis (Table 3). Nevertheless, this analysis is limited by the lack of standard indication for HBO.

According to multivariable regression analysis, elevated blood bicarbonate level (*p* = 0.032) and older age (*p* = 0.038) were associated with reduced incidence rate of globus pallidus necrosis. The inverse association between serum bicarbonate concentration and globus pallidus necrosis was not surprising, because those patients with globus pallidus necrosis also suffered from severer medical complications that could induce a greater degree of metabolic acidosis. Nevertheless, there was no clear explanation for the reciprocal relationship between age and globus pallidus necrosis.

## 5. Conclusions

Although patients who had attempted charcoal burning suicide had a low mortality rate (3.0%), globus pallidus necrosis was not uncommon (59.7%) in this population. Acute myocardial injury and neuropsychiatric symptoms were associated with an increased risk of globus pallidus necrosis. On the other hands, higher blood bicarbonate levels and older age were associated with a decreased risk of globus pallidus necrosis. Therefore, prompt diagnosis of risk factors and the use of HBO could possibly alter the clinical course of the disease and prevent the development of serious neurologic complications. Nevertheless, this study was limited by its retrospective nature, small sample size, and short follow-up duration. Furthermore, inclusion of patients was based on cerebral imaging, but there was no standard for this. This could have introduced bias. Further studies are warranted.

## Figures and Tables

**Table 1 ijerph-16-04426-t001:** Clinical data of patients who had attempted charcoal burning suicide, stratified by either presence or absence of globus pallidus necrosis (*n* = 67).

Variables	All Patients (*n* = 67)	Patients with Globus Pallidus Necrosis (*n* = 40)	Patients without Globus Pallidus Necrosis (*n* = 27)	*p* Value
Demographics				
Age, year	36.8 ± 11.1	34.5 ± 9.6	40.1 ± 12.4	0.044 *
Male, *n* (%)	45 (67.2)	25 (62.5)	20 (74.1)	0.428
Hypertension, *n* (%)	7 (10.4)	1 (2.5)	6 (22.2)	0.015 *
Diabetes mellitus, *n* (%)	6 (9.0)	2 (5.0)	4 (14.8)	0.211
Hepatitis B or C virus carrier, *n* (%)	6 (9.0)	3 (7.5)	3 (11.1)	0.679
Liver cirrhosis	1 (1.5)	0 (0)	1 (3.7)	0.403
Smoking habit, *n* (%)	37 (55.2)	21 (52.5)	16 (59.3)	0.796
Chronic alcohol consumption, *n* (%)	33 (49.3)	19 (47.5)	14 (51.9)	0.885
Past suicide history, *n* (%)	13 (19.4)	7 (17.5)	6 (22.2)	0.647
Mood disorder, *n* (%)	38 (56.7)	23 (57.5)	15 (55.6)	0.670
Adjustment disorder, *n* (%)	25 (37.3)	14 (35.0)	11 (40.7)	0.797
Clinical manifestations				
Systolic blood pressure, mmHg	115.6 ± 32.4	113.0 ± 31.8	119.4 ± 23.7	0.403
Heart rate, per minute	97.6 ± 23.4	101.9 ± 21.0	91.2 ± 21.3	0.066
Fever, *n* (%)	23 (34.3)	19 (47.5)	4 (14.8)	0.008 **
Acute respiratory failure, *n* (%)	27 (40.3)	20 (50.0)	7 (25.9)	0.075
Acute myocardial infarction, *n* (%)	27 (40.3)	21 (52.5)	6 (22.2)	0.022 *
Acute hepatic injury, *n* (%)	29 (43.3)	21 (52.5)	8 (29.6)	0.081
Acute kidney injury, *n* (%)	29 (43.3)	20 (50.0)	9 (33.3)	0.214
Acute rhabdomyolysis, *n* (%)	27 (40.3)	21 (52.5)	6 (22.2)	0.022 *
Neuropsychiatric symptoms, *n* (%)	34 (50.7)	28 (70.0)	6 (22.2)	<0.001 ***
Stroke, *n* (%)	8 (11.9)	7 (17.5)	1 (3.7)	0.103
Shock, *n* (%)	7 (10.4)	5 (12.5)	2 (7.4)	0.693
Out-of-hospital cardiac arrest, *n* (%)	2 (3.0)	2 (22.2)	0 (0)	0.512
Laboratory findings (normal range)				
Carboxyhemoglobin, % (nonsmoker 0.5–1.5, smoker 4–9)	21.8 ± 19.3	18.4 ± 3.2	26.7 ± 3.5	0.086
Blood urea nitrogen, mg/dL (6–21)	18 (10.0, 20.9)	19.9 (10.6, 23.3)	15 (9.0, 20.9)	0.164
Creatinine, mg/dL (M 0.64–1.27, F 0.44–1.03)	1.20 (0.8, 1.6)	1.4 (0.9, 2.1)	0.9 (0.8, 1.5)	0.022 *
Calcium, mg/dL (7.9–9.9)	8.1 ± 0.5	8.0 ± 0.6	8.2 ± 0.4	0.111
Phosphate, mg/dL (2.4–4.7)	3.3 ± 0.7	3.4 ± 0.8	3.3 ± 0.5	0.722
Sodium, mEq/L (134–148)	141.6 ± 3.1	141.8 ± 3.3	141.4 ± 2.9	0.613
Potassium, mEq/L (3.6–5.0)	4.2 ± 0.8	4.3 ± 0.8	4.1 ± 0.7	0.279
Albumin, g/dL (3.5–5.5)	3.4 ± 0.6	3.4 ± 0.4	3.4 ± 0.8	0.988
Aspartate transaminase, U/L (≤34)	75.0 (30.0, 196.1)	119.0 (55.3, 206.8)	39.0 (21.0, 150.0)	0.001 **
Alanine transaminase, U/L (≤36)	58.0 (26.5, 102.0)	77.0 (41.0, 104.0)	42.0 (17.3, 98.3)	0.080
Alkaline phosphatase, U/L (36–122)	60.2 ± 30.6	48.5 ± 3.7	76.3 ± 14.5	0.101
Bilirubin total mg/dL (≤1.3)	0.7 ± 0.3	0.8 ± 0.2	0.6 ± 0.4	0.479
Troponin I, ng/mL (<0.4)	0.9 (0.2, 2.8)	1.7 (0.6, 5.9)	0.1 (0.0, 0.8)	<0.001 ***
Creatine kinase, U/L (M 20–200, F 20–180)	2029.0 (238.0, 7227.0)	4610.0 (504.5, 13704.0)	303.0 (100.3, 2164.0)	0.020 *
Creatine kinase-MB, U/L (0.6–6.3)	11.9 (3.7, 28.5)	19.2 (8.8, 30.5)	4.1 (0.8, 12.5)	0.014 *
Myoglobin, ng/mL (M 17.4–105.7, F 14.3–65.8)	370.7 (180.6, 386.7)	386.7 (96.7, 1023.2)	41.8 (19.9, 166.1)	0.061
Hemoglobin, g/dL (M 13.5–17.5, F 12–16)	14.7 ± 2.2	14.6 ± 2.4	14.8 ± 2.0	0.800
White blood cell count, 10^3^/uL (M 3.9–10.6, F 3.5–11.0)	15.982 ± 8.064	16.592 ± 1.241	15.077 ± 1.624	0.455
Polymorphonuclear neutrophil, % (42–77)	81.6 ± 13.1	82.7 ± 13.7	80.0 ± 12.2	0.417
Platelet cell count, 10^3^/uL (150–400)	227.9 ± 71.2	228.7 ± 68.9	226.8 ± 75.7	0.917
C-reactive Protein, mg/L (<5)	29.2 (15.1, 74.2)	29.2 (13.8, 80.3)	27.6 (14.8 81.8)	0.897
Arterial blood gas				
pH (M 7.34–7.44, F 7.35–7.45)	7.37 ± 0.11	7.35 ± 0.13	7.40 ± 0.07	0.061
Partial pressure of carbon dioxide, mmHg (M 35–45, F 32–42)	35.1 ± 6.7	34.5 ± 6.4	36.0 ± 7.1	0.375
Partial pressure of oxygen, mmHg (75–100)	241.1 ± 146.4	235.7 ± 25.4	249.0 ± 24.0	0.706
Bicarbonate, mmol/L (M 22–26, F 20–24)	20.3 ± 4.5	19.1 ± 4.5	22.1 ± 4.1	0.007 **
Treatment and outcomes				
Oxygen therapy using non-rebreather facemask, *n* (%)	67 (100.0)	40 (100.0)	27 (100.0)	1.000
Hyperbaric oxygen therapy, *n* (%)	28 (41.8)	12 (30.0)	16 (59.3)	0.024 *
Mortality, *n* (%)	2 (3.0)	2 (22.2)	0	0.512

Note: F, female; M, male. * *p* < 0.05, ** *p* < 0.01, *** *p* < 0.001. Continuous variables are expressed as means ± standard deviations whereas categorical variables are expressed as numbers (percentages). Non-normal distribution data were presented as median (interquartile range).

**Table 2 ijerph-16-04426-t002:** Univariable logistic regression analysis of predictors for globus pallidus necrosis (*n* = 67).

Variable	Odds Ratio (95% Confidence Interval)	*p* Value
Acute myocardial injury (yes)	3.9 (1.3–11.6)	0.016 *
Neuropsychiatric symptoms (yes)	8.2 (2.6–25.3)	<0.001 ***
Bicarbonate level (each increment of 1 mmol/L)	0.8 (0.7–1.0)	0.010 *
Age (each increase of 1 year)	1.0 (0.9–1.0)	0.050 *
Acute rhabdomyolysis (yes)	3.9 (1.3–11.6)	0.016 *
Hyperbaric oxygen therapy (yes)	0.3 (0.1–0.8)	0.019 *
Creatinine level (each increment of 1 mg/dL)	2.8 (1.0–7.6)	0.044 *
Fever (yes)	5.2 (1.5–17.8)	0.009 **
Hypertension (yes)	0.1 (0.0–0.8)	0.030 *
Aspartate transaminase (each increment of 1 U/L)	1.0 (1.0–1.0)	0.021 *

Note: * *p* < 0.05, ** *p* < 0.01, *** *p* < 0.001.

**Table 3 ijerph-16-04426-t003:** Multivariable logistic regression analysis of predictors for globus pallidus necrosis (*n* = 67).

Variable	Odds Ratio (95% Confidence Interval)	*p* Value
Acute myocardial injury (yes)	4.6 (1.1–18.9)	0.034 *
Neuropsychiatric symptoms (yes)	8.0 (2.0–31.4)	0.003 **
Bicarbonate level (each increment of 1 mmol/L)	0.8 (0.7–1.0)	0.032 *
Age (each increment of 1 year)	0.9 (0.9–1.0)	0.038 *

Note: * *p* < 0.05, ** *p* < 0.01, *** *p* < 0.001.

**Table 4 ijerph-16-04426-t004:** Comparison of outcomes between current and published studies (sample size ≥5) from different geographic areas.

Study	Year	Country	Source of Carbon Monoxide	Sample Size	Incidence Rate of Globus Pallidus Necrosis, %	Mortality Rate, %
Current study	2019	Taiwan	Charcoal burning	67	59.7	3.0%
Jeon et al. [11]	2018	Korea	Not mentioned	387	19.9	Not mentioned
Moon et al. [12]	2018	Korea	Charcoal burning	128	18.7	0%
Kim et al. [10]	2017	Korea	Not mentioned	7	100.0	Not mentioned
Chen et al. [13]	2013	Taiwan	Charcoal burning	47	25.5	Not mentioned
Chen et al. [14]	2013	Taiwan	Not mentioned	30	53.0	Not mentioned
Chen et al. [15]	2005	China	Not mentioned	46	82.1 (magnetic resonance imaging); 43.2 (computed tomography)	Not mentioned
Durak et al. [16]	2005	Turkey	Not mentioned	8	37.5	Not mentioned
Kim et al. [17]	2003	Korea	Not mentioned	5	100.0	Not mentioned
O’Donnell et al. [4]	1999	UK	Not mentioned	19	63.2	21.1%
Matsushita et al. [18]	1996	Japan	Not mentioned	13	38.4	Not mentioned
Jones et al. [19]	1994	USA	Not mentioned	19	36.8	1.0%
Chang et al. [20]	1992	Korea	Not mentioned	15	60.0	Not mentioned

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
