# Peer review of "Incidence Rate and Predictors of Globus Pallidus Necrosis after Charcoal Burning Suicide"

_ijerph, 2019, doi:10.3390/ijerph16224426_

Round 1
Reviewer 1 Report
Title: Incidence rate and predictors of globus pallidus necrosis after charcoal burning suicide
Charcoal burning suicide is an important research topic. In this manuscript, the relationship between charcoal burning suicide patients and globus pallidus necrosis was investigated.
67 patients with charcoal burning suicide attempts were studied with their clinical information. The number of the subject is not small in consideration of the importance of the events.
The findings that patients with globus pallidus necrosis were younger and they had less hypetention are interesting. The amounts of the data of patients were listed well in tables. Analysis on variables seems to be conducted well. The statistical analysis seems to be conducted well. The percentage of globus pallidus necrosis patients among charcoal burning suicide attempt patients is interesting data.
I think this manuscript deserves to be discussed with other scientists.
However, there are several grammar errors in this manuscript.
Line 122: All patients received receive –
Line 156-Line 160: Please check grammar errors. Try to clarify the meaning of the sentence. L158 “may due to”
Line 209: elevated blood blood
Any graphical presentation of the data can improve communications of these data.
Author Response
Title: Incidence rate and predictors of globus pallidus necrosis after charcoal burning suicide
Charcoal burning suicide is an important research topic. In this manuscript, the relationship between charcoal burning suicide patients and globus pallidus necrosis was investigated.
67 patients with charcoal burning suicide attempts were studied with their clinical information. The number of the subject is not small in consideration of the importance of the events.
The findings that patients with globus pallidus necrosis were younger and they had less hypertension are interesting. The amounts of the data of patients were listed well in tables. Analysis on variables seems to be conducted well. The statistical analysis seems to be conducted well. The percentage of globus pallidus necrosis patients among charcoal burning suicide attempt patients is interesting data.
I think this manuscript deserves to be discussed with other scientists.
Response: Thank you for reviewing our manuscript
However, there are several grammar errors in this manuscript.
Line 122: All patients received receive –
Response: Thank you for your comment. The grammar error has been corrected.
Line 156-Line 160: Please check grammar errors. Try to clarify the meaning of the sentence. L158 “may due to”
Response: Thank you for your comment. The grammar errors have been corrected.
Increasing trend in charcoal burning suicide was noted in Asia countries. Mass media reporting plays an important role for the increased incidence of charcoal burning suicide. The higher incidence rate is believed to be due to traditional perception of the ethnic Chinese group who believe that a person should die with a complete body. The media portrayal of charcoal burning suicide as painless, easy, and no body disfigurement contributes to rapid spread of news in the era of globalization.
Line 209: elevated blood blood
Response: Thank you for your comment. The grammar error has been corrected.
Any graphical presentation of the data can improve communications of these data.
Response: Thank you for your comment. We apologize for there is no graphical presentation in this article.
Reviewer 2 Report
The authors studies patient after charcoal burning suicide attempt, and compared those with and those without globus pallidus necrosis.
The study does add some valuable data about this population to the literature, there are however some major points to be addressed.
The most important limitation is certainly the fact, that the authors seemingly simply analysed all parameters available, and reported where they found any significant results. Results should therefore be interpreted with extreme caution, and this should explicitly stated in the manuscript. The current statement "this study was limited by its retrospective nature" does not adequately reflect this, as the limitation is not the retrospective nature per se, but the way the data was analysed.
Another, not that severe, limitation is the fact that only patients with cerebral imaging were included. Authors should elaborate on how decisions on imaging were made, and if that could have introduced bias. Please also report how many patients (who would otherwise have met inclusion criteria) were excluded because of lack of imaging.
SPECIFIC COMMENTS:
Throughout the manuscript, replace univariate with univariable, and multivariate with multivariable.
Throughout the manuscript, please report odds ratios and the CIs only to one decimal place.
Results and first column of Table 1: In a cohort of 67 patients, it is not meaningful to report proportions to a degree of 0.1% (0.1% of 67 patients is 0.067 patients). Please round to full percentages.
Please state whether COHb reported was measured at admission (I would guess so), and which proportion of patients received O2 therapy prior to admission (as this greatly affects COHb levels).
Results, page 3, line 122: remove "receive". Please elaborate on the indications or SOP, on which decision for HBO treatment was based. Could this have introduced bias?
Please explain how "neuropsychiatric symptoms" were defined in your study.
In the last paragraph of page 3, authors state that according to regression analysis, AMI, neuropsychiatric symptoms, bicarbonate level, and age were risk factors for globus pallidus necrosis. However, only AMI and neuropsychiatric symptoms are reported to have odds ratios >1, whereas the other factors have ORs <1. This means that increased bicarb levels and higher age were associated with LOWER risk of globus pallidus necrosis (as the authors correctly state on page 7, in the last paragraph of the discussion). Please clafify the sentence on page 3, as those were not "risk factors for globus pallidus necrosis", but on the contrary.
Discussion, page 6, line 172: replace "was no difference" with "was not different"
In the conclusion, authors mix "risk" and "odds". They interpret the odds ratios as if they were risk ratios, which is not correct. I would recommend to remove the sentence beginning in line 220 with the words "In other words..." altogether.
Author Response
The authors studies patient after charcoal burning suicide attempt, and compared those with and those without globus pallidus necrosis.
The study does add some valuable data about this population to the literature, there are however some major points to be addressed.
The most important limitation is certainly the fact, that the authors seemingly simply analysed all parameters available, and reported where they found any significant results. Results should therefore be interpreted with extreme caution, and this should explicitly stated in the manuscript. The current statement "this study was limited by its retrospective nature" does not adequately reflect this, as the limitation is not the retrospective nature per se, but the way the data was analysed.
Another, not that severe, limitation is the fact that only patients with cerebral imaging were included. Authors should elaborate on how decisions on imaging were made, and if that could have introduced bias. Please also report how many patients (who would otherwise have met inclusion criteria) were excluded because of lack of imaging.
Response: Thank you for the comments.
We apologize for there was no routine use of brain imaging to aid in the clinical management of patients with charcoal burning suicide attempt. According to our medical records, only 67 out of 126 patients received brain image analysis. Furthermore, the indications for performing brain imaging were so diverse, such as neuropsychiatric symptoms, stroke, fever, or simply request by the family member, etc. This limitation has been stressed in Conclusion section.
Nevertheless, this study was limited by its retrospective nature, small sample size, short follow-up duration and lack of routine cerebral imaging. Further studies are warranted.
SPECIFIC COMMENTS:
Throughout the manuscript, replace univariate with univariable, and multivariate with multivariable.
Response: Thank you for your comment. The inappropriate terms have been replaced.
Throughout the manuscript, please report odds ratios and the CIs only to one decimal place.
Response: Thank you for your comment. The decimals have been rounded to one place.
Results and first column of Table 1: In a cohort of 67 patients, it is not meaningful to report proportions to a degree of 0.1% (0.1% of 67 patients is 0.067 patients). Please round to full percentages.
Response: Thank you for your comment. The proportions have been rounded to full percentages.
Please state whether COHb reported was measured at admission (I would guess so), and which proportion of patients received O2 therapy prior to admission (as this greatly affects COHb levels).
Response: Thank you for your comment. The carboxyhemoglobin levels were measured at admission. None of the patients received oxygen therapy prior to admission.
Results, page 3, line 122: remove "receive". Please elaborate on the indications or SOP, on which decision for HBO treatment was based. Could this have introduced bias?
Response: Thank you for your comments. The grammar error has been corrected.
In the Materials and method section, we have added
2.2. Oxygen therapy
There was no standard indication for hyperbaric oxygen (HBO) therapy. Nevertheless, the absolute contraindication for HBO was an untreated pneumothorax. Moreover, the relative contraindications comprised asthma, chronic obstructive pulmonary disease, seizure, fever, active infection, active malignancy, claustrophobia, pregnancy, and patients who need intensive medical care.
In the Discussion section, we have added
Nevertheless, this analysis is limited by the lack of standard indication for HBO.
Please explain how "neuropsychiatric symptoms" were defined in your study.
Response: Thank you for your comment. Neuropsychiatric symptoms were defined as new onset of cognitive and personality changes, dementia, psychosis, parkinsonism, amnesia, depression and incontinence after charcoal burning suicide.
In the last paragraph of page 3, authors state that according to regression analysis, AMI, neuropsychiatric symptoms, bicarbonate level, and age were risk factors for globus pallidus necrosis. However, only AMI and neuropsychiatric symptoms are reported to have odds ratios >1, whereas the other factors have ORs <1. This means that increased bicarb levels and higher age were associated with LOWER risk of globus pallidus necrosis (as the authors correctly state on page 7, in the last paragraph of the discussion). Please clarify the sentence on page 3, as those were not "risk factors for globus pallidus necrosis", but on the contrary.
Response: Thank you for your comment. The inappropriate sentences have been revised.
In a multivariable regression model (Table 3), it was disclosed that acute myocardial injury (odds ratio 4.6, confidence interval 1.1–18.9, P = 0.034) and neuropsychiatric symptoms (odds ratio 8.0, confidence interval 2.0–31.4, P = 0.003) were significant risk factors for globus pallidus necrosis. In contrast, bicarbonate level (odds ratio 0.8, confidence interval 0.7–1.0, P = 0.032), and age (odds ratio 0.9, confidence interval 0.9–1.0, P = 0.038) were protective factors associated with lower risk of globus pallidus necrosis.
Discussion, page 6, line 172: replace "was no difference" with "was not different"
Response: Thank you for your comment. The grammar error has been corrected.
In the conclusion, authors mix "risk" and "odds". They interpret the odds ratios as if they were risk ratios, which is not correct. I would recommend to remove the sentence beginning in line 220 with the words "In other words..." altogether.
Response: Thank you for your comment. The inappropriate sentences have been deleted.
Reviewer 3 Report
Dear Authors,
Article "Incidence rate and predictors of globus pallidus necrosis after charcoal burning suicide"
The article is very interesting and well set up. Its objective is the study of predictors of globus pallidus necrosis resulted from carbon monoxide poisoning after charcoal burning suicide. The research involved a total of 67 patients with charcoal burning suicide attempt, recruited and stratified into two subgroups based on either presence (n=40) or absence (n=27) of globus pallidus necrosis.
The points of strength are multiple: it is: well written, well structured, and very interesting, because there is almost no literature on the same object; the research answer is clear; the research design is well set up; the analyses are appropriate and correct; results emphasized the role of acute myocardial injury which seems to increase fold risk of globus pallidus necrosis.
The weaknesses concern the number of participants, which is too small and therefore the study does not allow a generalization of results.
Furthermore, the article could be improved if:
in the “Introduction” and in the “Discussion” considered better the literature on relationships between other diseases or traumas that cause hypoxia and globus pallidus necrosis and hypothesized why it is specifically important to analyze this cohort, indicating the strategic aims of this group of subjects; In the “Conclusion” the authors indicated the clinic implications of this discover, e.g. the use of Hyperbaric Oxygen Therapy after the suicide attempt; Charcoal burning suicide is a serious public health problem in Asia and is the first case of suicide in many countries, it would be important to give indications as to the possibility of disseminating among the population information about the risks involved in a suicide attempt using this method, because fear of disability can act as a deterrent.
Author Response
Dear Authors,
Article "Incidence rate and predictors of globus pallidus necrosis after charcoal burning suicide"
The article is very interesting and well set up. Its objective is the study of predictors of globus pallidus necrosis resulted from carbon monoxide poisoning after charcoal burning suicide. The research involved a total of 67 patients with charcoal burning suicide attempt, recruited and stratified into two subgroups based on either presence (n=40) or absence (n=27) of globus pallidus necrosis.
The points of strength are multiple: it is: well written, well structured, and very interesting, because there is almost no literature on the same object; the research answer is clear; the research design is well set up; the analyses are appropriate and correct; results emphasized the role of acute myocardial injury which seems to increase fold risk of globus pallidus necrosis.
Response: Thank you for reviewing our manuscript
The weaknesses concern the number of participants, which is too small and therefore the study does not allow a generalization of results.
Response: Thank you for your comments. This weakness has been stressed in Conclusion section. Nevertheless, this study was limited by its retrospective nature, small sample size, short follow-up duration and lack of routine cerebral imaging.
Furthermore, the article could be improved if:
in the “Introduction” and in the “Discussion” considered better the literature on relationships between other diseases or traumas that cause hypoxia and globus pallidus necrosis and hypothesized why it is specifically important to analyze this cohort, indicating the strategic aims of this group of subjects;
Response: Thank you for your comment.This important point has been stressed in the Discussion section.
There are some case reports noted about the correlation with acute myocardial injury and globus pallidus necrosis [25, 31, 32]. Only one registry-based cohort study showed that patients with CO poisoning have a higher risk of major adverse cardiovascular events [33]. The incidence rate of acute myocardial injury was 40.3% in our study (Table 1). Furthermore, patient with globus pallidus necrosis suffered higher rate of acute myocardial injury than patients without globus pallidus necrosis (52.5% versus 22.2%, P = 0.022). To our knowledge, 200 times greater affinity of carbon monoxide to hemoglobin then oxygen, so human body would suffer from hypoxia damage in the patients’ suicide by charcoal burning. The organs affected by carbon monoxide the most are central nervous system and the myocardium, which organs need oxygen the most. The patients with globus pallidus necrosis represented that affected more by hypoxia, so these patients would be in companied with myocardium injury. Also, oxygen delivery decreasing would make global oxygen demand and myocardial contractility increased that induced myocardial infarction in patients with underlying coronary artery disease [34]. Hypertension appeared to be a protective factor of globus pallidus necrosis in our study. While the elevation of blood pressure noted, the increased blood perfusion for all organs especially brain would be noted. Therefore, less hypoxia episode as well as globus pallidus necrosis will be noted in hypertension status. In support of this conception, a forensic pathology study [35] also indicated that globus pallidus necrosis is merelya consequence of cerebral hypoxia–ischemia from a wide variety of etiologies such as drug overdose, post-anesthesia, hemolytic uremic syndrome, acute renal failure, metabolic disorders, viral infections, porphyria and cerebral arteriosclerosis, delayed hanging and wasp stings, and is not specific for any kind of injury.
In the “Conclusion” the authors indicated the clinic implications of this discover, e.g. the use of Hyperbaric Oxygen Therapy after the suicide attempt; Charcoal burning suicide is a serious public health problem in Asia and is the first case of suicide in many countries, it would be important to give indications as to the possibility of disseminating among the population information about the risks involved in a suicide attempt using this method, because fear of disability can act as a deterrent.
Response: Thank you for your comment. The Conclusion section has been expanded.
Although patients with charcoal burning suicide had low mortality rate (3.0%), globus pallidus necrosis was not uncommon (59.7%) in this population. Acute myocardial injury and neuropsychiatric symptoms were associated with increased risk of globus pallidus necrosis. On the other hands, higher blood bicarbonate level and older age were associated with decreased risk of globus pallidus necrosis. Therefore, prompt diagnosis of risk factors and the use of HBO could possibly alter the clinical course of the disease and prevent the development of serious neurologic complications. Nevertheless, this study was limited by its retrospective nature, small sample size, short follow-up duration and lack of routine cerebral imaging. Further studies are warranted.
Round 2
Reviewer 2 Report
The authors addressed all the reviewer's comments in their response and overall strongly improved their manuscript.
As a last change I would recommend changing "...and lack of routine cerebral imaging." in the second to last sentence of the conclusion to new sentences, something like "Inclusion of patients was based on cerebral imaging, but there was no standard for this. This could have introduced bias."
Author Response
The authors addressed all the reviewer's comments in their response and overall strongly improved their manuscript.
As a last change I would recommend changing "...and lack of routine cerebral imaging." in the second to last sentence of the conclusion to new sentences, something like "Inclusion of patients was based on cerebral imaging, but there was no standard for this. This could have introduced bias."
Response: Thank you for your comment. The sentences have been revised.
Nevertheless, this study was limited by its retrospective nature, small sample size, and short follow-up duration. Furthermore, inclusion of patients was based on cerebral imaging, but there was no standard for this. This could have introduced bias.